# Investigation of Flow Changes in Intracranial Vascular Disease Models Constructed with MRA Images

**DOI:** 10.3390/s22062302

**Published:** 2022-03-16

**Authors:** Jeong-Heon Kim, Ju-Yeon Jung, Yeong-Bae Lee, Chang-Ki Kang

**Affiliations:** 1Department of Radiological Science, College of Health Science, Gachon University, Incheon 21936, Korea; air2kim431@gachon.ac.kr; 2Department of Health Science, Gachon University Graduate School, Gachon University, Incheon 21936, Korea; 9955me@gachon.ac.kr; 3Department of Neurology, Gil Medical Center, Gachon University College of Medicine, Incheon 21565, Korea; yeongbaelee@gachon.ac.kr

**Keywords:** magnetic resonance angiography (MRA), cerebral stenosis and occlusion, vascular disease models, time-of-flight (TOF) and phase-contrast (PC) MRA, MRI-compatible flow delivery system

## Abstract

This study aimed to develop a magnetic resonance imaging (MRI)-compatible flow delivery system and individualized models of circle of Willis (CoW), which include 50% and 100% blockage in internal carotid artery (ICA50 and ICA100), and 100% blockage in vertebral artery (VA100). Images were obtained using 3D time-of-flight and phase-contrast magnetic resonance angiography (MRA) sequences, and changes in velocity and flow direction at CoW models were analyzed. For the ICA50 and VA100 models, the flow was similar to that of the normal model. For the ICA 50 model, it was found that 50% blockage did not affect cerebral blood flow. For the VA100 model, decreased flow in the posterior cerebral artery and a change to the flow direction in the posterior communicating artery were found. For the ICA100 model, particularly, decreased flow in the ipsilateral middle and anterior cerebral arteries and a change to the flow direction in the ipsilateral anterior cerebral artery of the CoW were found. These results demonstrated that the flow system with various CoW disease models tailored to individual characteristics could be used to predict stroke onset more quickly. For the ICA50 and VA100 models, the possibility of cerebral infarction was significantly lower. On the other hand, for the ICA100 model, there was a high possibility of decreased flow, which could lead to cerebral infarction.

## 1. Introduction

Stroke is one of the leading causes of death for any reason [1], along with ischemic heart disease and cancer, and the quality of life after a stroke is often lower than that after a myocardial infarction [2,3]. An ischemic stroke, or cerebral infarction, in particular, accounts for the majority of stroke events [4]. A cerebral infarction occurs when cerebral perfusion falls below a certain threshold due to thrombosis and/or embolism of the arterial vessels, and the severity of the infarction is greatly affected by the function of the circle of Willis (CoW). The CoW is an anastomotic system of arteries in which the anterior cerebral artery (ACA), middle cerebral artery (MCA), and posterior cerebral artery (PCA) connect in a circle through the anterior communicating artery (Acom) and posterior communicating artery (Pcom). Therefore, when one of any related or connected vessels is stenosed or occluded, the CoW serves to protect the brain from cerebrovascular disease by providing constant and regular blood flow through the other connected vessels [5,6,7]. Therefore, the ability of the CoW to maintain constant blood flow is crucial to the prevention and prognosis of stroke.

According to previous studies, however, the structure of the CoW varies significantly between individuals [8,9]. Because of these differences, the possibility of maintaining cerebral perfusion through collateral flow in the event of a cerebrovascular disease, such as stenosis or occlusion, in large cerebral vessels may also vary among individuals [9]. For this reason, when cerebrovascular disease occurs, the possibility and severity of cerebral infarction differ among patients, depending on the function of the CoW. This also makes it difficult to determine the prognosis of the disease. Therefore, more information is needed to be able to estimate and predict how vascular diseases affect cerebral blood flow (CBF), based on an individual’s cerebrovascular status, such as where the cerebrovascular disease occurs in a normal cerebrovascular system and when it progresses from mild to severe.

Although there have been several computational fluid dynamics (CFD) studies evaluating cerebral perfusion in a variety of cerebrovascular disease models, many have been limited to in vitro studies [10,11,12,13,14]. Researchers have developed a one-dimensional (1D) model of the CoW to measure the average rate of change in velocity for each vessel based on the occlusion of the internal carotid artery (ICA) or vertebral artery (VA) [10,11]. Other researchers have developed two-dimensional (2D) and three-dimensional (3D) models to measure the flow rate according to the degree of ICA stenosis [12,13,14]. CFD studies, however, cannot be considered fully consistent with the actual values, as they approximate blood flow by simplifying the equation through multiple assumptions. Furthermore, cerebral artery diseases often cause turbulence, a phenomenon that is not fully understood and is a potential source of evaluation errors [15]. 

Another in vitro study evaluated changes in the flow rate and pressure based on different degrees of stenosis and occlusion in the left ICA segment of the CoW [16]. However, the modeling was performed using averaged physiological data rather than measurements of actual human cerebral vessels, and did not consider any individual physiological specifications, such as curvature or location of an individual’s vessels. Individuals have different vessel diameters, and the Acom and Pcom in particular, which are involved in the collateral pathway, have the highest variance between individuals [17]. Vessel diameter is the most important physiological factor in determining the ability of collateral flow. Consequently, even the existing CoW model studies, accounting for individual physiological specificities, have difficulties in representing hemodynamic changes in real human cerebral vessels. Additionally, previous studies have manufactured 3D CoW models from the vessel images acquired from computed tomography (CT) and/or magnetic resonance angiography(MRA), but these have been used to examine focal blood flow using digital subtraction angiography and sonography [18,19]. Therefore, these studies simply observed the blood flow in a cerebrovascular model. Until now, no previous study has used individual vascular models to evaluate the hemodynamic properties of vascular diseases by modeling various diseases that can cause changes in blood flow. To this end, studies are needed to measure hemodynamic changes by modeling cerebrovascular diseases, reflecting the specificity of individual cerebral vessels.

Therefore, the present study aimed to develop a system for predicting changes in CBF due to cerebrovascular diseases by reflecting the individual characteristics of the cerebrovascular system, and to observe how the maintenance of CBF is affected when diseases occur. We developed a flow system compatible with the magnetic resonance imaging (MRI) system and analyzed changes in blood flow velocity and direction in the CoW resulting from various cerebrovascular disease models.

## 2. Materials and Methods

A cerebrovascular model including the CoW was constructed from MRA images obtained from a young, healthy subject. Models of stenosis and occlusion of the right ICA and occlusion of the right VA were created from the normal cerebrovascular model. Using the flow system along with time-of-flight (TOF) and phase-contrast (PC) MRA sequences, images were acquired which were used to analyze changes in blood flow velocity and direction in the CoW based on various cerebrovascular diseases.

### 2.1. Circle of Willis Phantom

The process of designing and fabricating a cerebrovascular model was performed using the cerebrovascular MRA images of a healthy subject, who had no known high blood pressure, heart disease, kidney disease, or peripheral vascular disease (Figure 1). Using TOF MRA images, a normal blood vessel model was developed as follows. The digital imaging and communications in medicine (DICOM) images obtained from MRI scanner were segmented into the vessels of interest, including the Pcom, ACA, MCA, PCA, ICA, VA, Acom, and basilar artery (BA), using the modeling software InVesalius 3.1.0 (Centro de, Tecnologia da Informação Renato Archer, Campinas, Brazil). The images were then extracted into stereolithography (STL) files. Using Meshmixer 3.5 (Autodesk, San Rafael, CA, USA), the extracted STL files were fine-tuned to provide adequate blood flow, which was converted from DICOM data for removing unnecessary blood vessels. The normal cerebrovascular model had a vessel wall thickness of 2 mm, and was shaped in the form of a tube. The target ICA and VA were partially or fully blocked to form the cerebrovascular disease models. In the right ICA stenosis model, the vessel in the petrous ICA was blocked by 50% (ICA50) (Figure 2). The right ICA occlusion model was created by blocking 100% of the vessel at the same location as the stenosis (ICA100). Similarly, the right VA occlusion model was created by 100% blockage of the vessel just before the confluence of the VAs (VA100).

The normal cerebrovascular model along with all three cerebrovascular disease models were printed using a 3DWox 2X (Sindoh, Seoul, Korea) 3D printer with a 0.4 mm nozzle and a fused deposition modeling (FDM) method. All models were printed with a 1.75 mm diameter polyactive acid (PLA) filament (Sindoh, Seoul, Korea). Additionally, 100% infill was applied to prevent water leakage and to withstand water pressure, and the tensile strength was maximized using the rectilinear infill pattern setting [20].

### 2.2. Flow System

To generate blood pulse waves and reasonable flow rates, a flow system was constructed, as shown in Figure 3. A peristaltic pump, N6-12L (Shenchen, Baoding, China), a quantitative pump capable of transferring accurately defined liquid capacity, was used to generate blood pulses corresponding to the normal heart rate range. Additionally, a linear pump (SHURflo, CA, USA) was used to compensate for the shortfall in achieving an actual blood flow rate of approximately 800 mL/min [4], and an air separator (Flamco, Almere, Netherlands) was installed to remove artifacts caused by air bubbles in the tube. All equipment except for the cerebrovascular phantoms and appropriate tubes were placed outside of the MRI scan room so that they were not affected by the magnetic field.

### 2.3. MRI Protocols

MRA images were obtained using a 3T MRI scanner (Magnetom Vida, Siemens, Berlin, Germany) with a 20-channel head coil. Three-dimensional TOF MRA was used to evaluate for the presence or absence of morphological abnormalities in the phantoms, while 3D PC MRA was used to measure changes in the blood flow rate and direction. The MR parameters, as follows, are listed in Table 1. For TOF MRA, repetition time (TR)/echo time (TE) = 30/4.16 ms, acquisition time (TA) = 7 min 36 s, slice thickness (SL) = 0.43 mm, flip angle (FA) = 15°, field of view (FOV) = 192 × 96 mm^2^, voxel size = 0.4 × 0.4 × 0.4 mm^3^, and generalized autocalibrating partial parallel acquisition (GRAPPA) acceleration factor = 2. For PC MRA, TR/TE = 49.45/5.94 ms, TA = 11 min 33 s, SL = 0.63 mm, FA = 8°, FOV = 162 × 136 mm^2^, voxel size = 0.6 × 0.6 × 0.6 mm^3^, and GRAPPA acceleration factor = 2. Velocity encoding (VENC) was applied equally to all 4 phantoms at 50 cm/s along three directions (anterior–posterior, AP; right–left, RL; and superior–inferior, SI) to measure the velocity and direction of the blood flow. The velocity was calculated using the sum of the vectors. The MRI scan time for each phantom model was less than an hour, and after each model was finished, the MRI experiments were conducted with the next model in turn.

### 2.4. Image Analysis

The maximum intensity projection (MIP) technique was applied to the MRA images using ImageJ software (National Institutes of Health, Bethesda, MD, USA), and the resulting images were used to identify the actual structure of the phantoms and the areas of the reduced flow due to stenosis or occlusion. 

Using the PC MRA images, the flow velocity in each vessel was measured from the MIP image of the magnitude summed images, which were calculated using the square root of the sum of squares of each intensity component obtained from the three-directional VENCs. In the MIP image, the region of interest (ROI) for each vessel was selected using the ImageJ software. For example, in the right ICA, the ROI was selected from the segment of normal vessel after the stenotic end of the ICA50.

The mean signal intensity for each ROI was measured and used to derive the velocity change in each cerebrovascular disease model compared to the normal model. The direction of blood flow was determined in the PCA and Pcom with an AP plane image, the ACA and MCA with an RL plane image, and the ICA, VA, and BA with an SI plane (called also through plane) image.

### 2.5. Methodology

An radio frequency (RF) head coil was selected for MRI image acquisition on which a phantom was located with water bottles for reference; RF pulses were then applied to acquire k-space data. Fourier transform was performed for the image reconstruction of acquired k-space data. These procedures are all the built-in functions in the MRI system. In this study, we selected 3D TOF and PC sequences and their appropriate imaging parameters were used, providing high signal noise ratio (SNR) and high resolution. In the data analysis, the reslice technique was applied to compare and analyze the views of interest in three directions using the ImageJ program to see each vessel more accurately and clearly, but no interpolation during reslicing. In order to compare the regions of interest, MIP technique was performed to make only blood vessels visible, and then signal intensities for each vessel were analyzed. In addition, in order to analyze the direction of the flow, the magnitude map and the phase map were synchronized to match the position information of each vessel, and then data analysis was performed in the same ROI. 

## 3. Results

The 3D TOF MRA images clearly showed the morphological information for each modeled vascular disease (Figure 4). For the normal model, flow was observed in all vessels except for the Acom. Flow was similarly observed in the ICA50 stenosis model, but that model showed a narrowed vessel segment in the area where stenosis was present. For the ICA100 occlusion model, no blood flow was observed in the area where the vessel was blocked, but normal flow was observed in the other vessels. Furthermore, the flow in the Acom, which was difficult to detect in the normal model, was identified in this model. For the VA100 occlusion model, no flow was observed in the right VA, where the occlusion was located. Blood flow was observed in both Pcoms for all models, but in the Acoms, blood flow was observed only in the ICA100 occlusion phantom.

The PC MRA images showed changes in blood flow rates and directions between the vascular disease models. All of the vessels in the normal and ICA50 had normal flow, but for the ICA100, the flow in the right ACA was opposite to that of the normal phantom (Figure 5, Figure 6, Figure 7 and Figure 8). In the right ACA, the normal lateral to medial flow was in three models, but it was reversed in the ICA100. Additionally, for the ICA100, the flow of the Acom was clearly visualized, and the flow in the right ACA was medial-to-lateral (Figure 5, Figure 6, Figure 7 and Figure 8). For the ICA100, the blood flow in the right ICA was no flow in the post-occlusion segment, while a proximal-to-distal flow was observed in the other three models (Figure 6 and Figure 8). For the VA100, both Pcoms showed reversed flow, from anterior to posterior, while the other three models had a posterior-to-anterior flow (Figure 7 and Figure 8). 

The signal intensities of each vessel on the PC MRA images are shown in Table 2. Additionally, Figure 8 summarizes the average changes in velocity and direction of each blood vessel in the cerebrovascular disease models compared to the normal model. For the ICA50, the flow velocity decreased to 95% (e.g., −5% of the normal flow) in the right ICA and increased to 113% in the right Pcom. For the ICA100, there was an evident difference between the velocities of the left and right ICAs. Compared to the normal model, the velocity was decreased to 18% in the right ICA, but increased to 147% in the left ICA; the right ACA was decreased to 48%, but the left ACA increased to 133%; and the right MCA was decreased to 46%, while that of the left MCA increased to 124%. Both Pcoms showed significant increases in velocity, 261% and 132% in the right and left, respectively. The velocity in both VAs and the BA increased, owing to the influence of ICA occlusion. The velocity increased to 119% in the right VA, 113% in the left VA, and 123% in the BA, resulting in an increased velocity of both PCAs. For the VA100, the velocity increased to 134% in the left VA and decreased to 95% in the BA, while the right and left Pcom values decreased to 78% and 51%, respectively.

## 4. Discussion

The present study aimed to create a flow delivery system compatible with the MRI system, construct various vascular disease models, such as stenosis and occlusion of a unilateral ICA and occlusion of a unilateral VA, from a normal cerebrovascular model with an intact CoW, and to analyze the hemodynamic changes in the acquired MRA images to investigate the effects of each disease model on the CBF. Predicting the consequences caused by cerebrovascular disease is difficult; thus, modeling the individualized cerebrovasculature and developing the flow systems are an important approach.

In the analysis of the hemodynamic changes for each cerebrovascular disease model, we found little difference in blood flow velocity between the normal and ICA50. In contrast, for the ICA100, the reduced blood flow to the right ACA and MCA was compensated by both the Pcom and Acom. Even though the primary vessel was blocked, flow still existed with a velocity equivalent to 48% and 46% in the ACA and MCA compared to the normal phantom, respectively. Additionally, for the VA100, flow compensation was achieved by the reversal of the normal blood flow in both Pcoms, from anterior to posterior. 

The cerebrovascular images of the normal model were similar to those of the MRA image of the subject. For the normal model, the blood flow in the subject’s Pcom was posterior to anterior. Blood flow in the Acom was observed in the MRA images of the subject, but not clearly identified in the normal model, especially at the VENC of 50 cm/s, possibly due to the low blood flow through the Acom. Therefore, the value for the analysis could not be identified in the normal model, meaning that changes in the signal intensity in the Acom were not further evaluated.

When the right ICA was occluded by 100%, the flow velocities of the ipsilateral ACA and MCA varied by 48% and 46%, respectively. These variations showed that ICA occlusion affected most of the anterior circulation. These results have already been validated in several CFD studies [14,21,22]. Conversely, VA occlusion was found to affect the posterior rather than anterior circulation.

Of the collateral arteries in the CoW, the blood flow in the Acom was more active in the ICA100, as it was found to be compensated by the contralateral to the ipsilateral ACA, which was directly affected by occlusion. However, for the ICA50 and VA100, the flow change in the Acom was too weak to measure, because the reduced amount of blood flow was too small compared to the normal model [13,23]. 

Conversely, flow changes in both Pcoms were clearly observed in all models. Even for the ICA50, which has a similar blood flow to the normal model, the blood flow in the right Pcom increased to 113%, and for the ICA100, it increased to 261%. These results are similar to those measured using MRI and transcranial Doppler ultrasonography in clinical studies [9,24]. For the ICA100, the contralateral (left) Pcom was also found to have a 132% increase in flow. This is due to the actively working contralateral Acom compensating for the flow rate toward the disease, indicating that the Pcom provides an important collateral pathway for flow in the anterior direction in the case of unilateral ICA stenosis or occlusion [25,26]. 

As shown in Figure 7, it was confirmed that the collateral flow flowed from the anterior to posterior in both Pcoms in the VA100. Flow compensation by both Pcoms and PCAs was achieved because the flow rate of the BA was reduced due to the right VA occlusion. These results are consistent with those of prior clinical studies [27]. This result suggests that unilateral VA occlusion has less of an effect on CBF changes than unilateral ICA occlusion, because right MCA perfusion was clearly insufficient in the ICA100 compared to the VA100. 

When determining the prognosis related to possible stroke of the subject, relatively constant cerebral perfusion was maintained in the ICA50 and VA100 when compared to the ICA100, indicating a low incidence of stroke. In contrast, despite the well-developed collateral flow, the ipsilateral MCA had 46% flow reduction in the ICA100. Previous in vivo studies have shown that if the ipsilateral MCA was less than 60% of normal, the possibility of acute/subacute infarction was high [28,29]. Therefore, this decrease can result in poor blood supply to the vasculature of the frontal, parietal, and temporal lobes, which may increase the risk of stroke in these areas. However, it cannot be asserted that infarction has occurred, and other additional clinical information should be considered. Nevertheless, there is a possibility that the significant flow decrease in ipsilateral MCA will cause infarction. This kind of evaluation, performed with multiple cerebrovascular disease models tailored to individual characteristics, may make the changes in CBF more predictable, and enable various simulations, which ultimately could effectively prevent cerebrovascular diseases.

Despite these important advantages, the present study has several limitations. In the process of creating cerebrovascular vessel phantoms, the ophthalmic, choroidal, and superior cerebellar arteries were not included. Although these arteries do not directly affect CoW as much as the Pcom and Acom, they are essential to increasing the accuracy of blood flow predictions, as they can even minimally affect blood flow distribution and changes. Additionally, the CBF autoregulation mechanism in the human body and the elastic mechanism of vessels were not considered in the process of creating cerebrovascular disease models. Moreover, to meet individual characteristics, an individual’s CBF needs to be carefully measured and applied to the models. Therefore, designing more precise vascular model by extracting more realistic blood vessel information through image processing, de-noise, and enhancement of an individual’s blood vessel MRA image is necessary to perform future studies to obtain more realistic and applicable data.

In further study, for the exact flow velocity evaluation in the model, it would be necessary to directly compare with phase contrast MRA of the subject. However, as a previous study has shown, measurements of the velocity and direction of communicating arteries were not easy and they could be obtained from ultrahigh magnetic field MRI [30]. This might be because the velocity of these arteries requires various conditions, such as high-sensitivity devices, high-resolution imaging, and/or optimized PC MRA parameters [30]. Therefore, it would be good to conduct experiments to measure the velocity and direction of small cerebral vessels using an ultrahigh field MRI system. In addition, it would be necessary to develop a system that can regulate the flow of each inlet of the phantom, corresponding to that of each ascending vessel to the in vivo human brain.

## 5. Conclusions

The present study developed a flow system compatible with MRI, and investigated the hemodynamic phenomena seen in vascular disease models implemented in a normal cerebrovascular model, the results of which were consistent with those of previous studies. In addition, the subject was cautiously predicted to have a relatively higher probability of stroke in the presence of unilateral ICA 100% occlusion than in the presence of unilateral ICA 50% stenosis or VA 100% occlusion. The combination of a more advanced flow system and cerebrovascular models of individual characteristics will be meaningful in that they can effectively predict vascular diseases through prognosis and prediction of ischemic stroke. It would also potentially facilitate our understanding and insight into the hemodynamics of cerebral vessels, and eventually have applications in future treatments and diagnoses. 

## Figures and Tables

**Figure 1 sensors-22-02302-f001:**
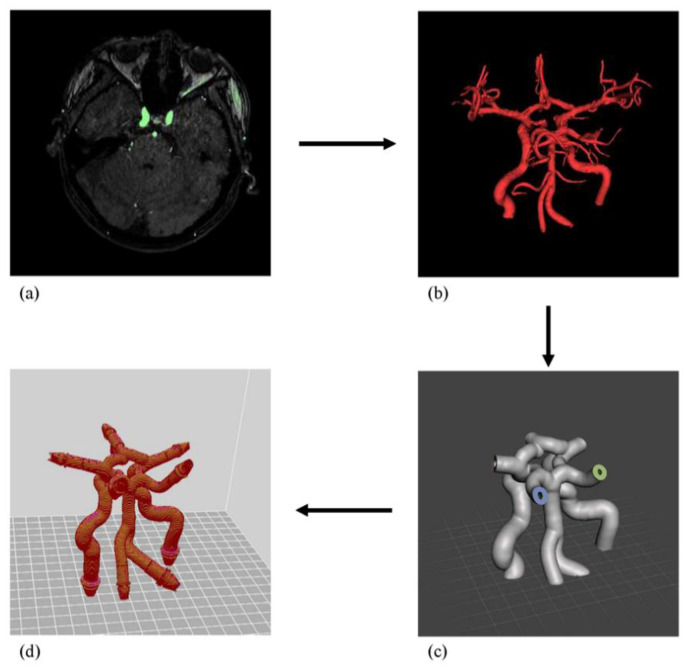
Workflow of three-dimensional (3D) vascular phantom manufactured from magnetic resonance angiography (MRA) images: (**a**) vessel segmentation from MRA images; (**b**) 3D model rendering; (**c**) refinement and modification for phantom; and (**d**) 3D model produced by a 3D printer. The black arrows indicate the workflow.

**Figure 2 sensors-22-02302-f002:**
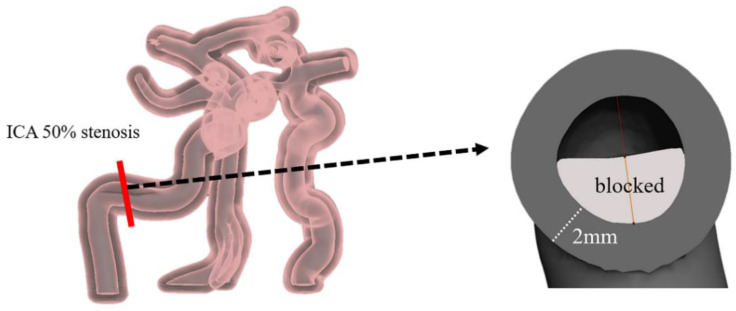
A cross-sectional image of the configuration of the stenosis in the right internal carotid artery (ICA) 50% stenosis model (ICA50). Note that the wall thickness and inner diameter of the vessel are 2 and 4.9 mm, respectively.

**Figure 3 sensors-22-02302-f003:**
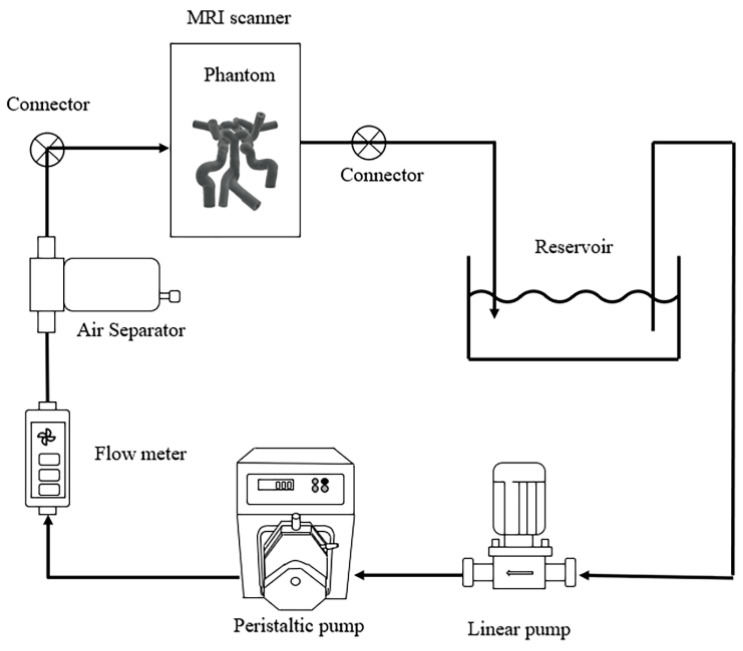
A schematic diagram for the experimental setup; the black arrows indicate the blood flow. Note that plastic connectors compatible with magnetic resonance imaging (MRI) are used to ensure proper extension of the tubes.

**Figure 4 sensors-22-02302-f004:**
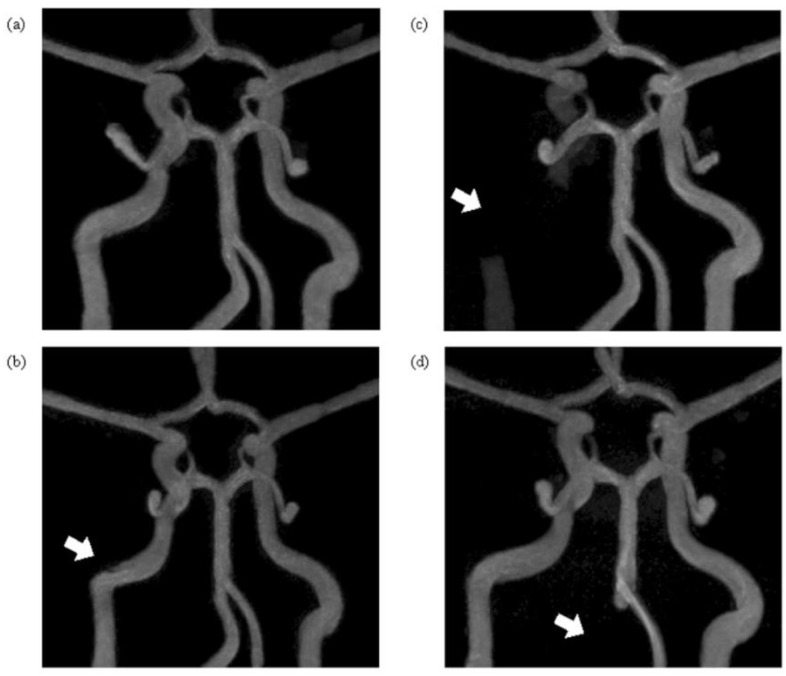
Time-of-flight (TOF) MRA maximum intensity projection (MIP) images of each vascular model: (**a**) normal phantom; (**b**) right ICA 50% stenosis phantom (ICA50); (**c**) right ICA 100% occlusion phantom (ICA100); and (**d**) right vertebral artery (VA) 100% occlusion phantom (VA100). The reconstructed TOF MRA images from each phantom were visualized using maximum intensity projection (MIP) in the coronal plane. The white arrows indicate the regions where cerebrovascular disease was modeled.

**Figure 5 sensors-22-02302-f005:**
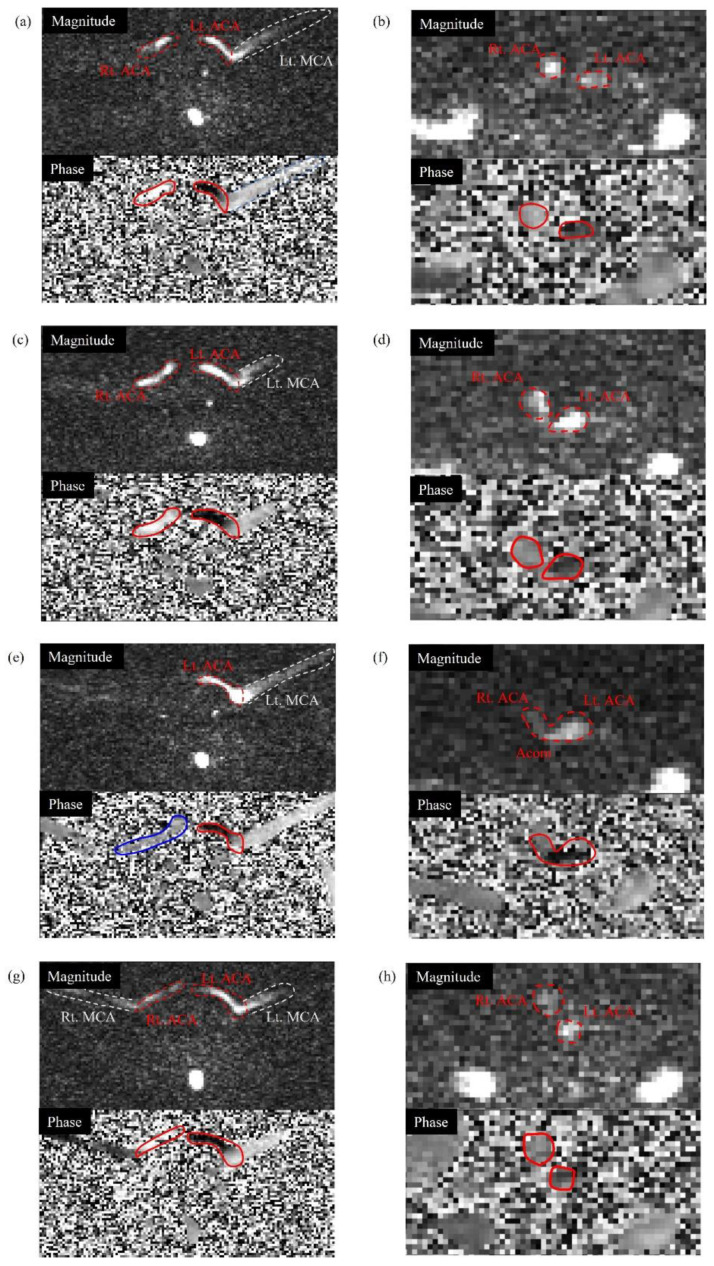
Representative slices for the analysis of the anterior cerebral artery (ACA) and anterior communicating artery (Acom) flow directions at velocity encoding (VENC) = 50 cm/s (magnitude and phase images on the top and bottom, respectively, and images of ACA and Acom on the left and right, respectively): (**a**,**b**) normal phantom; (**c**,**d**) right ICA 50% stenosis phantom (ICA50); (**e**,**f**) right ICA 100% occlusion phantom (ICA100); and (**g**,**h**) right VA 100% occlusion phantom (VA100). The phase images were obtained from a left–right encoding direction. The red lines represent a normal ACA and Acom flow, and the blue lines represent a reversed flow. The white dotted lines indicate the middle cerebral artery (MCA).

**Figure 6 sensors-22-02302-f006:**
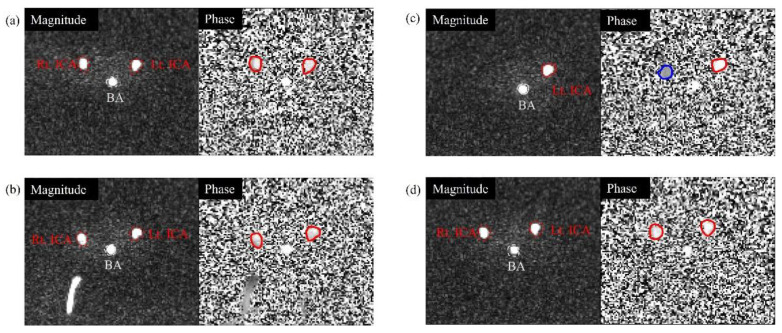
Representative slices for the analysis of the ICA flow directions at VENC = 50 cm/s (magnitude and phase images on the left and right, respectively): (**a**) normal phantom; (**b**) right ICA 50% stenosis phantom (ICA50); (**c**) right ICA 100% occlusion phantom (ICA100); and (**d**) right VA 100% occlusion phantom (VA100). The phase images were obtained from a superior–inferior encoding direction. The red lines represent the normal flow of the ICA, and the blue line represents no flow.

**Figure 7 sensors-22-02302-f007:**
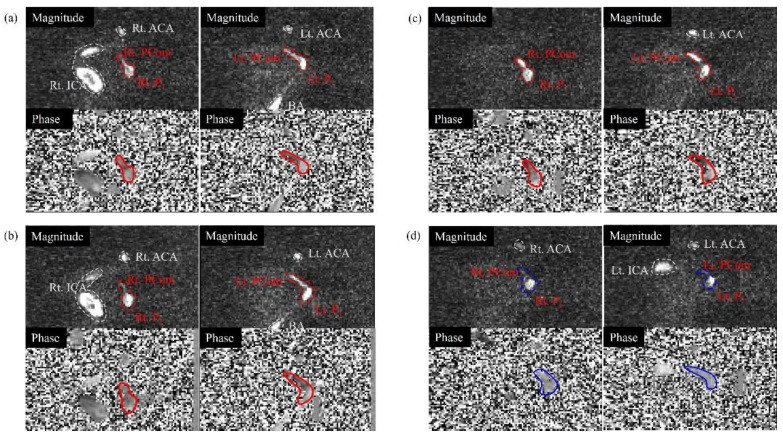
Representative slices for the analysis of the posterior communicating artery (Pcom) flow directions at VENC = 50 cm/s (magnitude and phase images on the top and bottom, respectively, and images of right and left vessels on the left and right, respectively): (**a**) normal phantom; (**b**) right ICA 50% stenosis phantom (ICA50); (**c**) right ICA 100% occlusion phantom (ICA100); and (**d**) right VA 100% occlusion phantom (VA100). The phase images were obtained from an anterior–posterior encoding direction. The red lines represent the normal flow of the Pcom, and the blue lines represent the reversed flow. The white dotted lines indicate the ICA.

**Figure 8 sensors-22-02302-f008:**
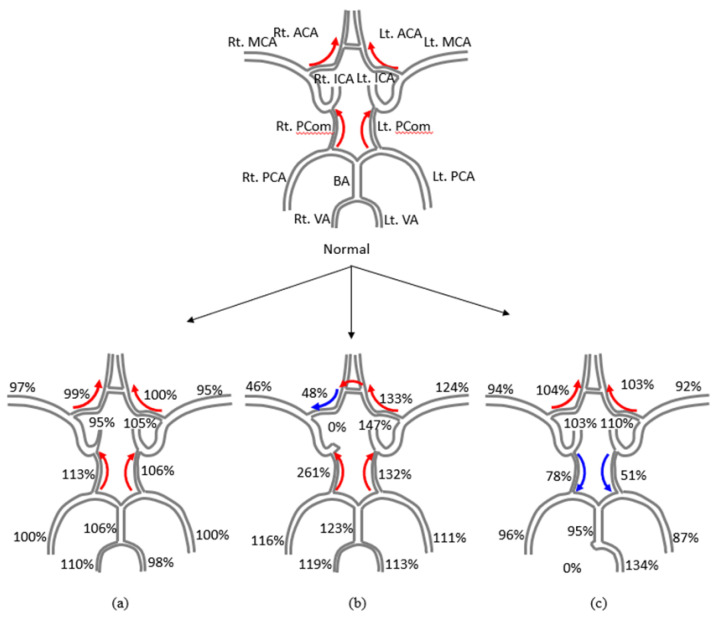
Velocity changes in each vessel of the cerebrovascular disease phantoms compared to the normal phantom: (**a**) right ICA 50% stenosis phantom (ICA50); (**b**) right ICA 100% occlusion phantom (ICA100); and (**c**) right VA 100% occlusion phantom (VA100). The red arrows indicate the same flow direction as the normal model and the blue arrow indicates a reversed flow direction compared to the normal model.

**Table 1 sensors-22-02302-t001:** Magnetic resonance (MR) imaging parameters.

Parameters	3D TOF	3D PC
TR (ms)	30	49.45
TE (ms)	4.16	5.94
TA (min:s)	7:36	11:33
FA (°)	15	8
FoV Read (mm)	192	162
FoV Phase (%)	50	84.4
Slice Thickness (mm)	0.43	0.63
Bandwidth (Hz/Px)	266	630
Voxel Size (mm)	0.4 × 0.4 × 0.4	0.6 × 0.6 × 0.6
GRAPPA	R = 2	R = 2
VENC (cm/s) and Gradient Directions	–	50 AP, RL, SI

Abbreviations: 3D, three-dimensional; AP, anteroposterior; FA, flip angle; FOV, field of view; GRAPPA, generalized autocalibrating partial parallel acquisition; MRA, magnetic resonance angiography; PC, phase-contrast; RL, right–left; SI, superior–inferior (through plane); TA, acquisition time; TE, echo time; TOF, time-of-flight; TR, repetition time; VENC, velocity encoding.

**Table 2 sensors-22-02302-t002:** Average signal intensities in the cerebral vessels of each phantom, corresponding to velocities.

	Normal Phantom			Rt ICA 50% Stenosis Phantom			Rt ICA 100% Occlusion Phantom			Rt VA 100% Occlusion Phantom		
Artery	Mean	SD	Max	Mean	SD	Max	Mean	SD	Max	Mean	SD	Max
Rt ICA	372	57	536	372	57	536	–	–	–	404	58	516
Lt ICA	436	44	578	436	44	578	612	73	758	457	42	543
Rt VA	486	98	703	486	98	703	527	96	712	–	–	–
Lt VA	497	30	528	497	30	528	571	134	761	675	121	878
BA	656	107	831	656	107	831	763	105	1012	528	108	732
Rt PCA	438	53	579	438	53	579	509	72	636	420	74	547
Lt PCA	421	66	578	421	66	578	467	100	715	366	95	706
Rt Pcom	116	30	174	116	30	174	269	53	387	80	35	167
Lt Pcom	227	60	333	227	60	333	283	38	349	109	24	165
Rt MCA	204	33	247	204	33	247	97	31	179	197	21	234
Lt MCA	179	34	267	179	34	267	226	53	363	167	39	266
Rt ACA	216	59	326	216	59	326	104	36	182	226	35	289
Lt ACA	231	43	314	231	43	314	308	34	355	239	38	323
Acom	–	–	–	–	–	–	147	33	193	–	–	–

Abbreviations: ACA, anterior cerebral artery; Acom, anterior communicating artery; BA, basilar artery; ICA, internal carotid artery; Lt, left; Max, maximum intensity; MCA, middle cerebral artery; PCA, posterior cerebral artery; Pcom, posterior communicating artery; Rt, right; SD, standard deviation; VA, vertebral artery.

## Data Availability

The data presented in this study are available on request from the corresponding author.

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
