# Peer review of "Investigation of Flow Changes in Intracranial Vascular Disease Models Constructed with MRA Images"

_sensors, 2022, doi:10.3390/s22062302_

Round 1

Reviewer 1 Report

The authors have provided an interesting study of the impact of moderate stenosis of the internal carotid artery and occlusions of either the internal carotid or vertebral artery. I was surprised to find that, in fact, there is little published on this seemingly obvious application of MR flow measurements under controlled conditions, given that suitable pumps and phantoms have been avaiable for about 20 years. 
Procedurally, the work is reasonably well done - the spatial resolutions of the images adequate, the segmentation and 3D printing and phantom + pump assembly are suitbly described and appropriate for the work undertaken. 

I have three substantial concerns with the work.
First, phase contrast MRA of the original subject is not described. It would be reassuring to see that the relative distributions of flow used in the phantom reflects that seen in the subject, and specifically the directions of flow in the communicating arteries corresponds to that seem in vivo. 

Second, the description in the Materials and Methods section regarding the extraction of velocity values centrers around signal intensities that are taken as square root of sum of squares. Not being fully up to date on how Siemens calculate their images, I  hesitate to say this is incorrect, but suggest to the reviewers to check whether the resulting images are truly velocity components (i.e. absolute value of phase map), or intensity weighted velocity components (i.e. magnitude of complex subtraction). The former would allow the authors to speak in terms of physical velocity values, which I believe would be much prefered by the community, rather than signal intensities. The use of intensities should not make a difference as to the statements regarding the sense of changes in flow velocity (i.e. increase / decrease), but I would hesitate before using such values in absolute flow measurements as they are clearly not in units of velocity (e.g. cm/s), and note that in cases of extreme difference in flows, the intensity weigthing would be likely to induce bias in the ratios of normal to stenotic / occlusive velocities. 

Third, the discussion is somewhat superficial, there is very little consideration of the prior literature and of how the present findings fit into the larger picture. This could touch on the evolution from angiographic depiction (Hartkamp and others) to flow measures, the impact of acute vs chornic pathologies for response mechanisms in the brain to flow obstruction (i.e. the model is acute, and doesn't take into account possible biological as opposed to hemodynamic adaptation), or prior works looking a relationships (and not always finding them) between flow measured with MRI and ischemic(like) changes in the brain. 

There is also one un-answered issue that should be addressed. How do the authors explain the retrograde flow into the post-occulsion stump of the occluded ICA in the ICA100 model? I suspect there is an aneurysm-like "loop" of flow, but that the returning blood volume is so saturatedthat it is not evidenced in the intensity images and may also not be accurately depicted in the phase maps. Or, is it a reason to doubt the values obtained in the other vessels?

The conclusions talk of infarct prediction, which I believe is not really supported by the present work.- relating the findings to prior works would help here...) . Are their in-vivo or simulation models that agree with the distributions you've observed? At the level of a "flow territory" how much would the perfusion be affected (i.e. the % reduction in flow in the MCA was? As a result of slow accumulation in atherosclerosis, is this chronically reducing an oversupply, or does it necessarily force areas of brain into misery perfusion or hypoxia? In and accute thrombotic event?

While the manuscript on the whole is quite well written, I have included a number of comments / edit suggestions.  

editting suggestions

throughout, the forms: "In ICA50", "In ICA100", "In VA100" and similar would probably read better as "For the ICA50", For the ICA100" or "For the VA100" ...

the decreased flow in                      decreased flow in 
and the changed flow                       and a change to the flow

Intro 
L56 done to evaluate                       evaluating
    in various cerebrovascular             in a variety of cerebrovascular

L69 the study was modeled using averaged physiological data rather than modeling from 
       the modelling was performed using averaged physiological data rather than measurements of

Materials and Methods

L110 what does "STL files were fine-tuned to provide adequate blood flow." mean? it seems an unusual feature of Autodesk.
L111 a vessel thickness of 2mm            a vessel wall thickness of 2mm 
L117 of both VAs                          of the VAs
L152-9 given that the parameters are listed in the table, they could be skipped in the text 
            were the base sequence of the time of flight and phase contrast acquisitions flow compensated on all axes? 
            wasn't a VENC = 0 used for referencing all velocity components, or a negative VENC image for each component acquired?

L177-178 does "intensity component" = the signal intensity image produced for a given velocity component encoding, the complex subtraction (respect to what as a reference or negative VENC image), or the phase difference (respect to what reference or negative VENC image)? if the intensity image, it should be remembered that the value scaling is not strictly defined by the VENC, and thus the absolute velocities may be misleading, and phase values would be better. Looking only at relative velocities, there should not be a problem. 

L181 flow completely passed through        at the point of maximum narrowing? 
                                                ref ... 
L189 & 190 the flow was                    flow was

L194 even in the normal model              in the normal model

L205 rates and directions in the           rates and directions betweenn the

L208 I suspect you mean the proximal ACA as above the confluence it is difficult to image flow coming caudally.

L210-211 , which originally had a lateral-to-medial flow,  delete, this is already stated for the other models.

Might it be possible to illustrate the positions of these slices on the MIPS of Fiugre 4? 

Could Figures 5-8 be unified in a single page (rotated), the captions are so similar it doesn't 

Table 2 - why signal intensities - this is a study of velocity mapping!

Discussion 
L290  images of the subjects                 wasnt't there just one subject imaged? 

L320   clearly insufficient                  how was this assessed? are there clinical guidelines on how much flow is insufficient? 

ref 12 - why upper case for names?

ref 19 - s.V.                               S.V. ? 

Reviewer 2 Report

The paper is devoted to an important problem related to the ischemic heart disease and stroke. Several specific comments are bellow:

Major comments:

  1. ABSTRACT: I suggest to modify the abstract to include notes to the proposed methodology and results achieved

  1. Section 1 INTRODUCTION and 2 MATERIALS AND METHODS: These section are well organized and they present a very clear description of the flow system and its modelling. I suggest to specify here also appropriate mathematical methods for image processing, de-noising, and enhancement. Associated references should be added, for instance:

[1] Langari B., at al.: Edge-Guided Image Gap Interpolation Using Multi-scale Transformation, IEEE Transaction on Image Processing,  25(9): 4394-4405, 2016

[2] Khanal A., at al:  Evolutionary origins of the blood vascular system and endothelium, Journal of Thrombosis and Haemostasis,11(Suppl.1): 46–66, 2013

  1. Section 2.2 FLOW SYSTEM. Was the flow system presented in Fig. 3 used by authors and experiments evaluated by authors? Which results were achieved?

  1. Section 2.4 IMAGE ANALYSIS: The own contribution of authors should be specified here. Were some software components created by authors? Or evaluation parameters were selected only? How they were specified?

  1. I suggest to add section 3 METHODOLOGY to specify the proposed  method of data acquisition, sensors (cameras) used, the frame rate, image resolution, data processing, software tools, hardware, and evaluation time.

  1. Section 3 RESULTS (may be Section 4 after renumbering):  The proposed mathematical method of image components detection (in Figs 5-8) should be better and more clearly described. Which methods of image processing and enhancement were used?

  1. Section 4 DISCUSSION (may be Section 5 after renumbering):  Was some kind of comparison of models with real vascular system done? The possibility of the use of results in the clinical practice should be mentioned in the final section.

Minor comments:

  1. Columns of Tables 1 and 2 should be narrower to have tables not so wide

Round 2

Reviewer 2 Report

Most comments were answered and I suggest the publication of the paper.